# Electrocardiogram (ECG)-Based User Authentication Using Deep Learning Algorithms

**DOI:** 10.3390/diagnostics13030439

**Published:** 2023-01-25

**Authors:** Vibhav Agrawal, Mehdi Hazratifard, Haytham Elmiligi, Fayez Gebali

**Affiliations:** Department of Electrical and Computer Engineering, University of Victoria, Victoria, BC V8P 5C2, Canada

**Keywords:** user authentication, Electrocardiogram (ECG), deep learning algorithms, LSTM and PTB database, telehealth system, CNN and LSTM training and validation

## Abstract

Personal authentication security is an essential area of research in privacy and cybersecurity. For individual verification, fingerprint and facial recognition have proved particularly useful. However, such technologies have flaws such as fingerprint fabrication and external impediments. Different AI-based technologies have been proposed to overcome forging or impersonating authentication concerns. Electrocardiogram (ECG)-based user authentication has recently attracted considerable curiosity from researchers. The Electrocardiogram is among the most reliable advanced techniques for authentication since, unlike other biometrics, it confirms that the individual is real and alive. This study utilizes a user authentication system based on electrocardiography (ECG) signals using deep learning algorithms. The ECG data are collected from users to create a unique biometric profile for each individual. The proposed methodology utilizes Convolutional Neural Networks (CNNs) and Long Short-Term Memory (LSTM) to analyze the ECG data. The CNNs are trained to extract features from the ECG data, while the LSTM networks are used to model the temporal dependencies in the data. The evaluation of the performance of the proposed system is conducted through experiments. It demonstrates that it effectively identifies users based on their ECG data, achieving high accuracy rates. The suggested techniques obtained an overall accuracy of 98.34% for CNN and 99.69% for LSTM using the Physikalisch–Technische Bundesanstalt (PTB) database. Overall, the proposed system offers a secure and convenient method for user authentication using ECG data and deep learning algorithms. The approach has the potential to provide a secure and convenient method for user authentication in various applications.

## 1. Introduction

One-dimensional biomedical signals, such as the electrocardiogram (ECG) or electroencephalogram (EEG), can be used as biometric characteristics [1]. ECG traces reveal information on electric cardiac transmission and are utilized to identify specific users. ECGs are strongly tied to the individual features of each person’s heart. It is well recognized that each person’s heart has distinct physiological features due to genetic variances. Such minor variations impact the cardiac beats and may be seen in each user’s ECG pulse. Many methods have been implemented to benefit from these variations while using Vital signs for user verification [2].

To verify identity in a platform, biometric technology uses biological information. Facial images, fingerprinting, eyes, vision, palm veins, speech, and even the form of the ears are some of the most popular data sources [3]. Moreover, several challenges must be overcome by Electrocardiography biometric technology, such as (i) elevated pairwise variations brought on by pulse signs and conducted actions; (ii) a distinct lack of studies demonstrating the discrimination ability in massive data, (iii) protracted modifications in a person’s ECG’s features. Particularly uncontrolled collection conditions can significantly degrade the efficacy of ECG biometric devices. Figure 1a shows the position of the electrode to record an ECG signal, while Figure 1b shows an ideal ECG signal with no problems. It shows the traditional method in which the electrical impulses that enable the heart to pump are recorded using chest-mounted electrodes. In this new era of technology, a wearable electrogram consists of different sensing systems used to collect the ECG data from the patient wrists or fingertips.

The Electrocardiogram (ECG) is made up of three major segments that correspond to various cardiac operations: atrial depolarization (P wave), ventricular repolarization (T wave), and ventricular depolarization (QRS complex). There are three ECG feature types: hybrid, non-fiducial, and fiducial. Fiducial features retrieve discrete-time properties from the Electrocardiography [4], which are calculated as time frames, pulse width, angles, and dynamical intervals based on the distinctive locations inside the ECG waveform. Non-fiducial characteristics alter the feature points using transformation functions, whereas hybrid characteristics integrate fiducial and non-fiducial techniques. Researchers consider fiducial techniques because they only employ readings of fiducial markers of an Electrocardiogram as characteristics inside the temporal domain [5]. The Electrocardiogram is a diagnostic technique that is frequently used to examine the muscular and electrical functions of the cardiovascular system by monitoring the heart rate and activity.

Figure 2 presents the Telehealth system connected through cloud services with local hospitals, clinics, emergency services, and patients at home. Healthcare practitioners may check a person’s health over time from a distance thanks to wearable technology, unobtrusive sensing, and telemedicine, which can assess symptoms and early illness warning indicators. This would enable improved management through the early detection and surveillance of illness signs, thus eliminating the requirement for face-to-face interaction [6,7]. Several challenges have been identified in using telehealth systems in the last decades. Some of the main problems include the following:Security concerns: Telehealth systems handle sensitive medical information and personal data, so they must be secure to protect against unauthorized access and data breaches. This can be challenging, especially given the increasing sophistication of cyber threats.Interoperability issues: Telehealth systems often need to integrate with many other systems, such as electronic health record systems, insurance providers, and lab systems. Ensuring interoperability between these systems can be complex and time consuming.Limited user acceptance: Some people may be hesitant to use telehealth systems due to concerns about their privacy, the perceived invasiveness of the technology, or a lack of understanding about how the systems work.Limited accessibility: Telehealth systems may not be accessible to everyone, particularly in areas with limited broadband connectivity or other infrastructure challenges.Regulation and compliance issues: Telehealth systems are subject to a wide range of regulations and requirements, which can be complex and time consuming.Technical challenges: Telehealth systems can be complex and require specialized infrastructure and support to function effectively. Ensuring that the systems are reliable, scalable, and user friendly can be challenging.

The authentication method has new possibilities because of the recent advancements in deep learning (DL) and machine learning (ML) classification algorithms. The branch of artificial intelligence known as ML uses training data to create statistical formulas, most often regression (predictions) systems and decision modeling techniques (e.g., categorization and feature identification) [8,9,10,11]. One of the many uses for ML is the analysis of pictures, audio, videos, and ECG information. DL has been applied to the top ML algorithms to enhance their accuracy. Deep neural networks perform better in application scenarios than conventional categorization methods. They effectively resolve complex issues because they can evaluate various input features. The findings have demonstrated that the DL neural models can likely offer greater prediction accuracy and efficient performance [12,13].

Previous reviews related to work have used traditional image processing, ML, and DL methods for user authentication using Electrocardiogram data. Nabil et al. [14] proposed a novel biometric authentication system EDITH that identifies individuals using electrocardiography (ECG) data. This technique can provide a safe and dependable method of authenticating people based on their distinct ECG patterns. The key benefit of EDITH is that it analyzes ECG data using deep learning algorithms, providing a high level of precision in identifying individuals. Furthermore, because ECG signals are distinctive to each individual, it is difficult for someone to imitate another person using this technology. The necessity for specialized hardware, such as ECG sensors, to gather and interpret ECG data is one possible area for improvement with EDITH. However, as this technology becomes more popular and accessible, deploying EDITH in several scenarios may become more practicable. A system developed by [15] to evaluate data from the Internet of Things (IoT) devices in a healthcare environment is the Event-driven IoT architecture for data analysis of trustworthy healthcare applications employing complicated event processing. This system analyzes data from IoT devices, such as sensors and wearable devices, using an event-driven architecture and advanced event-processing algorithms. One of the system’s key benefits is its capacity to analyze massive volumes of data in real time, allowing for rapid and accurate examination of healthcare data. The application of complicated event-processing algorithms also enables the discovery of patterns and correlations in data, which may be beneficial for detecting trends and predicting outcomes. This system’s capacity to grow to accommodate many IoT devices is another advantage, making it suitable for application in various healthcare settings. Furthermore, the event-driven architecture integrates many data sources, such as electronic medical records and sensor data, to give a more comprehensive perspective of a patient’s health.

Hamza et al. [16] provides a two-step process, consisting of data extraction and classification, for conducting person authentication using ECG data. Three additional forms of feature indices, ZCR, and entropy, are combined in the initial stage. The support vector machines (SVM) have been utilized for the categorization scheme in stage two. They evaluated the model on two public benchmark databases, obtaining the highest precision and accuracy values. Salloum et al. [17] suggested using RNNs to create a feasible option for classification and identification issues in ECG-based identification. Compared to earlier techniques that utilized the ECG-ID or MITDB databases, this work has shown that long short-term memory recurrent neural networks provide an efficient approach to Electrocardiography biometric authentication and identification. Prakash et al. [18] provides BAED (a protected biometric authentication system based on ECG signals and deep learning techniques), which is a biometric identification technique that recognizes persons using electrocardiography (ECG) signals and deep learning techniques. BAED’s usage of ECG signals, which are unique to each individual and give a high level of security, is one of its key benefits. BAED’s deep learning algorithms allow great accuracy in identifying individuals based on their ECG data. One advantage of BAED is that it employs secure protocols to preserve ECG data and prevent unwanted access. The system guarantees that the system is safe and that people’s privacy is respected.

Zaghouani et al. [19] proposed a security-based framework to safeguard the private information sent across ECG communication links, sometimes referred to as “unreliable channels.” In the suggested technique, the ECG characteristics are hidden via linear prediction encoding rather than being sent to the recipient end. Therefore, the cryptographic keys are produced on the recipient’s end rather than being transmitted. The proposed method showed outstanding results by keeping the patient’s privacy in safe hands. Karpinski et al. [20] created and validated a novel approach relying on auto-encoder neural network models to detect and eliminate ECG pulse abnormalities. The processed and standardized ECG data are sent into the auto-encoder algorithm. The input and reconstructed data are evaluated, and the root-mean-square loss is computed. The error function is employed to find inaccurate information. The research findings have been contrasted with previously created methods. The internet of things (IoT) elderly health monitoring system based on biological and behavioral indicators is a system developed to monitor the health of senior adults using biological and behavioral markers proposed by Hosseinzadeh et al. [21]. This system collects data on numerous indicators such as heart rate, blood pressure, and sleep habits using IoT devices such as sensors and wearable devices. It can provide real-time monitoring of a person’s health, allowing for quick action in the case of a health problem. The utilization of many indicators, including biological and behavioral markers, provides a complete picture of an individual’s health. Another advantage of this system is its capacity to remotely monitor older people, which is especially beneficial for those who live alone or have mobility concerns. This can offer caregivers and family members peace of mind while lowering the chance of falls or other incidents.

Hamza et al. [22] suggested combining various techniques to identify people. Their model focuses primarily on a mixture of attributes, including the prosodic and acoustic properties of every section of the ECG waveform. SVM is used to assess the effectiveness of the suggested strategy for recognizing people. The proposed system has an average accuracy of 92.5% with ECG-ID benchmark data, and with the MIT-BIHA collection, it has an accuracy rate of 98.6%. Belo et al. [23] have demonstrated the value of Deep Neural Networks for improving existing biometric technology. To enhance the outcomes of both the recognition (actual man) and verification (actual identification) procedures, the following two structures are proposed: Recurrent neural network (RNN) and Temporal convolutional neural network. Overall, the results demonstrate that the TCNN surpasses the Recurrent neural network, reaching nearly 100% and 90% efficiency for recognition and 0.1% and 2.2% equivalent margin of error for authenticating operations.

Lee et al. [24] used an ensemble of 2D CNN and LSTM ECG data to achieve personally identifiable information. During the first stage, noise reduction and standard variation correction were carried out. After contrasting and comparing one long short-term memory layer and two LSTM layers, the only one ECG with distortion eliminated was classified using two LSTM layers with considerably greater accuracy. The overall efficiency of every 2D CNN and LSTM was enhanced via ensemble learning, and the identification accuracy increased from 1.06% to 3.75% relative to a single framework. Kim et al. [25] presented a new long short-term memory Deep RNN design based on ECG classification and conducted an operational assessment for the algorithm using different databases. The outcomes demonstrate that the recommended methodology is more effective than other traditional approaches and surpasses them. The tests reveal that the suggested model outperforms the traditional long short-term memory technique in terms of classification precision and performance by achieving an F1 score of 0.99, and precision, recall, and accuracy of 99.8%.

Akeem et al. [26] proposed an improved Electrocardiography biometrics verification scheme for application. They employed a regression-based interpretable ML technique to establish the database bounds and obtain high-quality data for training. A collaborative regression analysis was then employed to create the benchmark functionality for every Electrocardiogram data entity (i.e., identification). The developed program’s authenticating performance was determined using a confusion matrix with the amgecg toolkit in MATLAB, which investigates two crucial variables. The recommended system shows high performance during testing. Labati et al. [27] proposed a new technique called Deep ECG for biometric identification based on an Electrocardiogram signal. This method can be used mainly for three bio-metric tasks: confined recognition, proof of identity, and periodical re-authentication. In order to obtain a selection of characteristics, a deep CNN network is used to evaluate groups of QRS waves that have been recovered using ECG data. Simple and binary templates are compared using Hamming distances and Euclidean techniques. The proposed technique achieves high performance and efficiency.

Biran et al. [28] present an automation system enabling person identification utilizing randomly chosen Electrocardiogram signals. The proposed approach is based on a mix of short-time Fourier transforms for separating frequency information and FMD for recognizing individual objects. Repeated measurements using this technology have shown good subject identification outcomes. The proposed architecture achieves an average of 96% accuracy in the evaluation process.

The research works in this part demonstrate the efforts to create various CNN designs to recognize people using ECG data. Still, handling such complicated data requires a precise approach. We suggest that the CNN and LSTM architecture creates efficient and precise identification using ECG signals. In order to obtain characteristics that allow for closed-set recognition, biometric identification, and recurring verification, this paper offers DL frameworks based on CNNs and LSTMs. The open access database from the Physionet dataset tests the suggested technique using quality metrics. The study’s main contributions are as follows:We illustrate the implementation of bidirectional Recurrent neural networks based on Long Short-Term Memory (LSTM) in addition to 1D-CNN for ECG authentication by achieving state-of-the-art accuracy.We perform data pre-processing to improve the performance of our DL models. To speed up the convergence of our model, the redundant values are deleted.We conduct thorough tests and exams on widely known baseline methods and compare our technique to state-of-the-art methods.We demonstrated that our models could be applied to ECG data obtained under various circumstances, delivering superior or equivalent reliability to the top-performing state-of-the-art techniques.

The remainder is arranged as follows: The majority of the ideas put forward in the research are discussed in Section 2. The proposed methodology that we utilized to evaluate the most popular approaches realistically is shown in Section 3. Section 4 explores the experimental use of deep learning neural networks in the field of ECG-based methods. Section 5 discusses future work and conclusions.

## 2. Materials and Methods

This section offers an in-depth analysis of the suggested technique with algorithms. The literature review above served as an inspiration for the suggested method in this part. We employed deep learning (DL) techniques to address every study question. Each DL model’s parameters and several layers have been altered to get the best accuracy. Figure 3 provides a visual representation of the preprocessing, training, and assessment phases of ECG identification. The suggested technique is based on deep learning algorithms that are trained and optimized utilizing a wide range of hyperparameters. The learning rates and biases in neural networks are optimized accordingly.

### 2.1. Convolution Neural Networks

The most sophisticated DL-based algorithms for learning robustness and automatically distinguishing features are convolution neural networks. Deep learning uses numerous convolutional layers to reflect learning features based on data. CNN has been used for various cognitive activities, including computer vision [29], natural language [30], and others. Many Convolutional networks, including Caffe-Net [31], Alex-Net [32], and VGG-Net [33], have been created for large-scale image recognition. Each phase of this convolution layers encoder collects the features from the input layer. A vector of a predetermined length is created by condensing the data. The database contains a wide range of inputs. Max Pooling, Convolution, and Dense layers are used to achieve the results.

The CNN is presented with a batch of ECG signals and their corresponding labels (permitted or denied). The CNN processes the ECG signals through a series of layers, starting with the input layer and ending with the output layer. Each layer consists of several units, which are connected to the units in the previous and next layers by weights. As CNN processes the ECG signals, it learns to recognize features characteristic of authenticated and not-authenticated signals. These features are learned through convolution, which involves sliding a kernel (a small matrix of weights) over the input data and computing dot products between the kernel and the input data at each position. The dot products are then passed through an activation function, which determines whether or not the unit should be activated (i.e., whether it should contribute to the output of the layer). As the CNN processes the ECG signals, the units’ weights are adjusted to minimize the difference between the predicted and true labels. This process, known as backpropagation, continues until the CNN has learned to classify the ECG signals accurately. Once the CNN has been trained, it can be used to classify new ECG signals and determine whether or not they are authenticated. This is achieved by presenting the CNN with the new ECG signal and using the output of the CNN to make a prediction

The design of the proposed CNN shown in Figure 4 was employed in this investigation to create the ECG feature patterns. It has five convolutional layers, with a max pooling layer, a fully connected layer, and a soft-max layer following each one. In all networks, the FC layer has the same design. Because local response standardization does not enhance the efficiency of our Electrocardiogram collection, but rather causes an increase in computation time and memory usage, all FC layers are provided with (ReLU) activation.

The CNN was configured with the following layer: Inputs → Convolution-1D (16 features, map size 7, relu) → Max Pool Layer(stride 3,2) → Convolution-1D (32 features, map size 5, relu) → Max Pool Layer(stride 3,2) → Convolution-1D (54 features, map size 5, relu) → Max Pool Layer(stride 3,2) → Convolution-1D (128 features, map size 7, relu) → Max Pool Layer(stride 3,2) → Convolution-1D (256 features, map size 7, relu) → Max Pool Layer(stride 3,2) → Convolution-1D (256 features, map size 8, relu) → Max Pool Layer(stride 3,2) → Flatten Layer → Dense (100 units) → SoftMax. The Adam optimization technique with categorical cross entropy produces better outcomes.

### 2.2. Long Short-Term Memory Networks

The long-short-term memory (DL) model for serial and time analysis is the most popular deep learning (LSTM) model, which solves the poor memory problem. With the aid of the gates referred to in its internal structure, which are employed to control the flow of information, LSTM solves this poor memory problem. The LSTM model works best for time series data, such as translating, meteorology, and voice recognition. Its operating method of gating carries forward the crucial data inside the lengthier dataset to produce precise and effective predictions [34].

The long-term dependence issue of a recurrent neural network was proposed to be resolved by a LSTM with architecture more sophisticated than a RNN [35]. A LSTM has an input gateway, forget gate, and outlet port to avoid data redundancy. Using the output value, the sigmoid transfer function returns a number between zero and one, indicating how much information there is. As a result, it can add or remove cell state information. A LSTM’s activation functions are the sigmoid [36] and Relu nonlinear activation functions [37]. In contrast with the forget gate, which controls whether previous data are removed from the cell state, the input gate controls whether new data are retained in the cell state. The output gate chooses which information from the cell state should be output in the meantime.

LSTM networks retain information from previous time steps in the data by using a particular unit called a memory cell, which is designed to retain information over an extended time. Each memory cell in a LSTM network has three gates: an input gate, an output gate, and a forget gate. The input gate determines which information from the current time step should be added to the memory cell. In contrast, the forget gate determines which information from the previous time step should be discarded. The output gate determines which information from the memory cell should be used to predict the current time step. Figure 5 shows the main components of the LSTM cell.

The information retained in the memory cell is combined with the input data at the current time step to produce an output, which is then passed to the next layer of the LSTM network. This process is repeated at each time step, allowing the LSTM network to retain information from previous time steps and use it to make predictions at future time steps. In the case of ECG data for authentication systems, the LSTM network would be trained to learn features from the ECG signals indicative of authenticated and not-authenticated signals. As the LSTM network processes the ECG signals, the memory cells would retain information about the characteristics of the signals, which would be used to make predictions about the authenticity of the signals at each time step.

Therefore, in this study, the proposed method considered the result at each time stamp rather than only collecting the concealed factor at the output. Model 2 in Figure 6 represents the LSTM model’s suggested design. Each time-output stamp from the LSTM cell is added together and then transmitted via a dropout layer. After that, a fully linked layer receives the outputs of the single hidden layer. Last but not least, the chance that an ECG sequence belongs to a person is determined using a softmax function. The candidate individual is chosen as the one whose likelihood is at its highest. The LSTM’s construction is shown in Figure 6.

A LSTM layer was added to the most recent algorithm implementation to check whether the results were noticeably better than those of other networks. The layer of the network was set up as follows: LSTM Layer (64 filters) → , Dropout (0.2 rates) → LSTM Layer (32 filters) → Dropout Layer (0.2 rates) → Flatten Layer → Dense Layers → SoftMax. Better results are obtained using the Adam optimization strategy with categorical cross-entropy.

### 2.3. Authentication System

The process of confirming and guaranteeing an object’s identity is known as authentication. Given the significance of data in the healthcare system, authentication can offer access control by determining whether a user’s credentials match the readily available records on the server. ML and DL provide the key to using evolving authentication. To reduce security barricades and deal with security issues, ML can be established. A key measure to prevent unwanted administrator privileges is to use ML to authenticate only those organizations, such as IoT devices, medical professionals, and patients, before allowing them to communicate with system resources. Deep learning classifiers calculate the resemblance of a dataset to every data classification process. Then, based on their definition of a probability measure, they select the category with the highest probability based on similarity. In contrast, classifier-based authentication models train exclusively using valid class samples. The first class will be chosen for the classifier in both scenarios in Table 1 because it has the highest probability value. In contrast, the authentication model requires a two-stage algorithm that follows the flowchart in Figure 7. The sample gathered from the individual is submitted to the trained classifier, which determines whether or not the predicted and claimed labels match. If the sample fits the claimed label within the limits of the threshold (θ), the system will identify the individual and authenticate it further; otherwise, the suggested classifier will reject the sample, and the entrance will be denied.

## 3. Data Collection and Simulation Using Deep Learning Algorithms

The ECG authentication system can be used in two modes: verification (authentication) or identification (recognition). The verification mode is used to validate an asserted identity. It is used to determine whether a person is whom he or she claims to be. This mode is used to authenticate a mobile user. The score is compared to a preset threshold during verification, and the claimed identity is approved if the score is more significant. The identification method is used to categorize and identify an unknown identity. It answers inquiries such as “who is this person?” and “is this person in the database?” This method is commonly used in cybercrime. During identification, the highest matching score is taken into account. ECG bio-metric design is divided into two stages: enrollment and authentication. Finally, the assessment criteria will be used to make a choice. Figure 7 presents the schematics of authentication methodology.

The ECG data are gathered, analyzed, and saved as a reference template throughout the enrollment phase. The ECG test sample is checked against the stored reference template(s) in the Authentication phase to ascertain the similarity or dissimilarity score provided by machine learning technique(s) against a predetermined threshold. Firstly, ECG signals are acquired by putting electrode leads on the person’s body (on the person) or through wearable devices (dry electrodes, off the person). After recording the ECG signals, they are pre-processed before feature extraction. Finally, the samples are saved in a feature database and categorized using classifiers such as decision trees, support vector machines, CNN, etc. The following sub-sections provide a detailed categorization of each component.

### 3.1. Data Collection

ECG signals are electrical signals that may be recorded by placing up to 12 electrodes (sensors) on a person’s chest and limbs. The Physikalisch-Technische Bundesanstalt (PTB) has one of the most extensive on-the-spot ECG databases. The data were obtained from the ECG-ID Database. The collection comprises 310 ECG recordings from 90 people. Figure 5 shows main components of ECG signal. Each recording includes: (i) ECG lead, recorded for 20 s and digitized at 500 Hz with 12-bit resolution across a notional ten mV range; (ii) 10 annotated beats (unaudited R- and T-wave peak annotations from an automatic detector); (iii) information (in the record’s .hea file) includes age, gender, and recording date. The records were acquired from volunteers (44 men and 46 women aged 13 to 75 years who were the author’s pupils, coworkers, and acquaintances). The number of recordings obtained for each person ranges from two (collected on one day) to twenty (collected over six months). The processed dataset has 31,000 rows and 202 columns. The raw ECG data, including high and low-frequency noise, are noisy. Each document includes both raw and filtered signals: Signal 0: ECG I (raw signal) Signal 1: ECG I filtered (filtered signal).

### 3.2. Feature Engineering

The process of obtaining relevant features from raw data that can be utilized to train a machine learning model is known as feature engineering. Feature engineering in the context of electrocardiogram (ECG) signals for telehealth authentication entails determining the most important and informative parts of the ECG signals for the job at hand. This may entail detecting certain waveform patterns, amplitudes, or frequency components that are indicative of a given physiological state or feature. This is a crucial action performed throughout the enrollment and authentication phases. This may entail feature extraction and selection, followed by a training classification based on the extracted features. Previous research on traditional machine learning algorithms has revealed certain flaws. They require a highly complicated foundation to construct the authentication mechanism. As a result, many characteristics will be retrieved from signals. Too many features (the curse of dimensionality) will need additional training time and memory space, thus impeding real-time implementation. Machine learning algorithms may also struggle with massive test datasets. This also has the issue of overfitting, which can lead to performance loss. As a result, traditional machine learning approaches that use static and hand-crafted features are time consuming and tedious. They can be replaced by deep learning approaches that can self-learn useful features from input ECG signals, thus providing a more straightforward framework for an authentication system.

The features were extracted from the .hea file using 200 Hz frequency samples, the average number of samples acquired per second. Patient, age, gender, RR, ECG mean, ECG standard deviation, ECG variance, ECG median, and ECG samples were the demographic and statistical information collected. However, following additional investigation, we only used the ECG raw data to detect QRS peaks. The raw EGC value was then employed as a predictor, with the patient as the target feature. The following is a detailed description of the feature engineering process:1All patients’ ECG records are loaded by the .hea file.2For each patient record, the ECG is loaded by Waveform Database Software Package (WFDB) library, and the signal is re-sampled in 200 Hz.3It is necessary to determine the groups of peaks that correlate with local signal maxima after re-sampling the signal. It must detect the positions of QRS peaks using the GQRS detection technique.4The Pearson correlation coefficient is calculated using the QRS average. An array of correlations is produced, and the eight highest values are picked.5Following that, the index CORR array signals are normalized and placed in a new array signal temp. Each array is associated with a column in the final data frame.

### 3.3. Feature Selection

There is no single collection of sufficient or dependable features for correctly categorizing every ECG signal under all situations. Some characteristics may discriminate effectively in some circumstances but not in others. Using several features and varied combinations can increase classification efficiency, but rigorous feature selection is also required to keep errors to a minimum [3]. A data frame represents the extracted characteristics listed above. The peaks were extracted using only the raw ECG data, as seen in Figure 5 below:

Its mean was calculated using an array of peaks, and its Pearson correlation was calculated using its mean. The last characteristic was the signal with the most excellent Pearson correlation. The entire procedure may be summarized as follows:1Load the ECG data file and extract the physical signal.2Resample the physical signal at 200 Hz: fs = 200.3Detect QRS sites in a single channel electrocardiogram. This is a straight transfer of the GQRS algorithm from the original WFDB package in terms of functionality. Accept physical or digital signals with known ADC gain and ADC zero.4Select identified peaks and adjust them to correspond with local signal maxima. Utilize a radius and a window size to average the array values and provide an array of the corrected peak indices to modify the set of identified peaks.5Normalize the index CORR array signals are normalized and place them in a new array signal temp. Each array is associated with a column in the final data frame.

To summarize, the entire procedure extracts the peaks with a specific frequency, which was set to 200, resamples them, and adjusts them to suit the maximum value. All values are placed in an 8-by-8 sub-array with the highest correlated values. The array values are then normalized between the lower and higher bounds. These are the last data records in the ptb.csv file. Overall, resampling the frequency of ECG data at 200 Hz using the WDBSP involves importing the data, applying any necessary preprocessing steps, and then using the resampling function to interpolate the data to the new sampling frequency. It is important to carefully consider the implications of resampling the data and to ensure that the resampled data are still representative of the original signal.

### 3.4. Hyperparameters and Loss Function

Hyperparameters and the loss function play an essential role in training deep learning models because they determine the model’s behavior and performance during training. By choosing appropriate hyperparameters and a suitable loss function, it is possible to improve the model’s ability to learn features from the ECG data and accurately classify the signals. For example, setting a more extensive learning rate may allow the model to learn more quickly but may also result in unstable training and overfitting of the training data. On the other hand, setting a lower learning rate may result in slower learning but may also improve the model’s generalization to new data. Similarly, choosing a larger batch size may speed up training but may also result in less accurate gradients, while choosing a smaller batch size may result in more accurate gradients but slower training. By carefully selecting the appropriate hyperparameters and loss function, it is possible to improve the model’s ability to learn features from the ECG data and accurately classify the signals for authentication purposes.

In order to produce an effective method for tackling the problem, this section explains how well these loss functions and hyperparameters are selected. The effectiveness of a DL model is determined by accuracy and loss. As DL models strive for the lowest error rate feasible, a system is more effective if the calculated loss is more minor than the actual calculated loss. In multi-class classification, we employ categorical cross-entropy to determine the average gap between expected cost, expected values, and loss measurement. The categorical cross-entropy is shown below Table 2. Weights can fast approach the local minimum during training by employing a flexible gradient descent mechanism. We chose Adams over other optimizers such as RMSProp [38] or SGD [39] to ensure the best loss reduction results, a superior learning process, optimal memory utilization, and implementation simplicity. The values of the hyper-parameters are shown as small learning rates (LR). We employed a batch size of 32 to prevent computational memory from becoming overloaded when transferring data over a network. Adam accomplished fast convergence more quickly and effectively. In order to see the response, each model has been trained over fifty periods with a predetermined period.
(1)$$L_{\left(CCE\right)}=\sum_{q=1}^{l}y_{q}*log\left({\hat{y}}_{q}\right)$$

## 4. Results and Discussion

The proposed CNN and LSTM were trained using raw ECG signals from the dataset. The outcomes of the validation and training are covered in this part. The reliability and size of this dataset were improved using several cleaning and feature extraction approaches. We utilized various sets of hyperparameters and reported those that provided state-of-the-art results. We utilized the Adam optimizer [40] with a learning rate of
$10\times 10^{-3}$, which gradually decreased to
$10\times 10^{-5}$ as the epochs increased to improve the metric results. We also utilized a mini-batch of 32 samples per batch and binary cross entropy loss [41] function to train our proposed architectures. Furthermore, the Softmax classifier was employed to provide the final probabilities after completing the training. The key benefit of implementing Softmax is the spectrum of outcome probabilities. The probability distribution would be zero to one, and the total of all probabilities was one. If the softmax function was employed in a multi-classification framework, then possibilities of every category were returned, with the target class having the highest chance [42]. While the experiment was operating, we trained our CNN and LSTM using the Keras API and a backend TensorFlow. In total, 80% of the training examples and 20% of the testing dataset was used to train the suggested architects. The hyperparameters in this design plan demonstrate that accuracy grew steadily over a short period as the number of epochs increased, stabilizing at a certain number. CNN displays the best accuracy of 96.8% with the least amount of loss. The training and validation loss for our CNN model is shown in Figure 8a,b during training. The LSTM algorithm worked admirably on the PTB dataset, achieving 95.6% validation accuracy. The obtained loss and accuracy curves for the suggested LSTM model are shown in Figure 8c,d.

### Evaluation Metrices

A standard tool for assessing how accurately a model might forecast a particular validation collection is the confusion matrix (CM). The CM includes comparable columns and rows that display the test dataset labels and the actual class. For each validation sample, the projected values show the proportion of accurate and inaccurate forecasts or categorizations. The number of adequately classified positive samples is known as True Positive, while the amount of successfully foreseen negative instances is known as Negative Cases. False Positives are forecasts in which the item was marked as positive but was not. False negatives are unfavorable outcomes that have a good appearance. Numerous metrics, including accuracy rate, recall, precision, F1 score, sensitivity, the area under the curve, and specificity, were used to assess the effectiveness of the AI-based algorithms. The following Equations (Equation 2)–(Equation 5) determined each model’s accuracy rate, recall, precision, F1 score, sensitivity, and specificity [43].
(2)$$Precision=\frac{TP}{TP+FP},$$
(3)$$Recall=\frac{TP}{TP+FN},$$
(4)$$Accuracy=\frac{TP+TN}{TP+TN+FP+FN},$$
(5)$$F1-Score=2\times \frac{Precision\times Recall}{Precision+Recall}.$$

The Receiver Operating characteristic (ROC) graph is a measurement tool for classifiers concerns. Essentially, it isolates the “signal” from the “background” by plotting the true positive rate (TPR) against the false positive rate (FPR) at different threshold levels. The capacity of a predictor to differentiate among classes is measured by the Area Under the Curve (AUC), which is used as a summarization of a Prediction model. A model having a higher AUC value means the model is efficient at predicting between FPR and TPR. The ROC curve for the proposed CNN and LSTM method is displayed in Figure 9 and Figure 10, respectively. The curve performed better the greater the value on the left. The ROC curve may determine how the TPR will vary as the FPR increases from 0 to 1. The FPR is zero when the threshold is stated as one but changes to one whenever the criterion is set as zero. The ROC curve is created using the TPR based on the modification of the FPR value. Table 3 shows comparison to other state-of-the-Art methods.

After building, training, and evaluating the LSTM and CNN models, a threshold value of 0.7 was defined to filter patients with the highest probability of being classified. From the resulting subset, the patient with the highest result was selected. We utilized the cross-validation approach to examine several thresholds and found that a threshold value of 0.7 outperformed other accuracies. Choosing thresholds greater than this value is more restrictive and can strengthen security, but in this case, the false positive rate increased, significantly reducing accuracy. Furthermore, utilizing numbers less than this might raise the false negative rate, which is unsatisfactory in such authentication systems. By choosing the threshold of the probabilities of the 90 patients for one measurement (INDEX-CNN), we obtained the index of the dataset that has the patient number. These thresholds were applied to the resulting array, creating a subset of probabilities greater than or equal to the defined threshold. Table 4 shows evaluation results of proposed CNN and LSTM models.

The performance of a convolutional neural network and a long short-term memory network for an authentication system using electrocardiogram (ECG) data will depend on several factors, including the data’s quality, the model’s complexity, and the training procedure. One key factor that can impact the performance of a CNN for ECG data analysis is the quality of the data. If the ECG data are of high quality and contain apparent, distinguishable features, the CNN will be better able to learn to recognize these features and classify the data accurately. However, if the data are noisy or contain artifacts, the CNN may need help extracting useful features and perform poorly.

The complexity of the CNN model can also impact its performance. A more complex model, with more layers and filters, may be able to extract more subtle features from the data, but it may also be more prone to overfitting. On the other hand, a simpler model may be less prone to overfitting and less capable of extracting nuanced features from the data. Similarly, the performance of a LSTM for ECG data analysis will depend on the quality of the data and the complexity of the model, as well as the training procedure. Properly tuning the hyperparameters of the LSTM, such as the number of memory cells and the learning rate, can help improve its performance. In general, both CNNs and LSTMs can provide good performance for ECG data analysis tasks, but the specific performance will depend on the specific data and the design of the model. It may be helpful to experiment with different model architectures and training procedures to find the combination that works best for a given dataset and task.

## 5. Conclusions and Future Work

The use of the Electrocardiogram for personal authentication is a relatively new area of research. According to research, the ECG signal is employed in the multimodal fields of emotion identification and medicine, among others. An ECG waveform is employed as a biometric and a helpful diagnostic. This article proposed a minimal intelligent system for identity verification using ECG data. This process utilizes neural network models to identify complicated QRS segments and carry actual user identity on these unprocessed QRS segments to simplify the architecture. In contrast with prior approaches, this research has shown that a CNN and LSTM-based RNN is more effective for Electrocardiography biometrics authentication and identification. The suggested technique directly feeds the CNN and LSTM with the ECG signals, eliminating the need for fiducial extraction of features or other characteristics such as oscillation and fractal components for both scenarios. A RNN-LSTM classifier is trained as a classification for the recognition issue, and for PTB datasets, 100% performance was attained. The suggested method’s adaptability to other aberrant cardiac disorders and various physiological situations requires a more thorough investigation. This research shows that Electrocardiography-based biometric authentication and identification are exciting applications for CNNs and LSTM-based RNNs.

Several potential approaches could be taken to enhance the security of telehealth systems by adopting electrocardiogram (ECG)-based deep learning authentication systems. Some potential strategies include:Improving the quality of the ECG data: One key factor that can impact the performance of an ECG-based deep learning authentication system is the quality of the data. Ensuring that the ECG data are of high quality and free from noise or artifacts can help improve the accuracy of the system.Developing more advanced deep learning models: Another potential approach is to develop more advanced deep learning models that can extract meaningful features from the ECG data. For example, researchers could explore using more complex model architectures, such as multi-modal models that combine data from multiple sources or using unsupervised or semi-supervised learning techniques to improve the system’s performance.Enhancing the security of the data transmission process: Telehealth systems rely on transmitting sensitive data over networks, which can be vulnerable to attacks. Adopting secure data transmission protocols, such as encryption and secure socket layers (SSL), can help protect the data as they are transmitted.Implementing multi-factor authentication: Another potential approach is to implement multi-factor authentication, which requires multiple forms of identification to access the system. In addition to using ECG data for authentication, this could include other biometric factors such as facial recognition or fingerprints or other forms of identification, such as passwords or security tokens.Regularly updating and testing the system: It is essential to regularly update and test the ECG-based deep learning authentication system to ensure that it continues to function effectively and provide adequate security. This could include periodic testing to identify and address potential vulnerabilities and regularly updating the system with the latest security patches and updates.

## Figures and Tables

**Figure 1 diagnostics-13-00439-f001:**
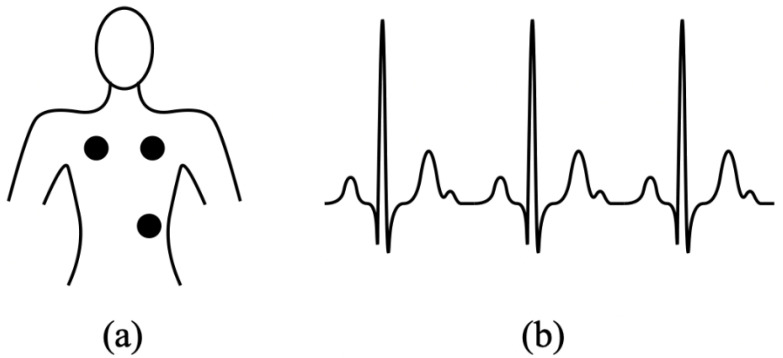
(**a**) Electrodes position for ECG recording (**b**) ideal ECG signal.

**Figure 2 diagnostics-13-00439-f002:**
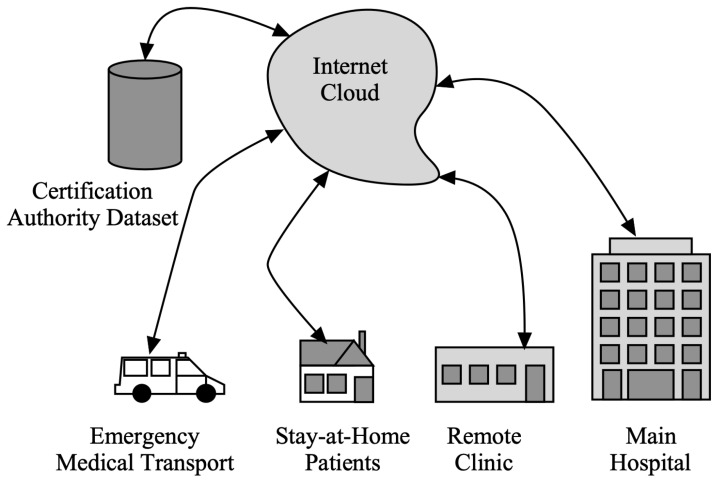
Telehealth system under consideration.

**Figure 3 diagnostics-13-00439-f003:**
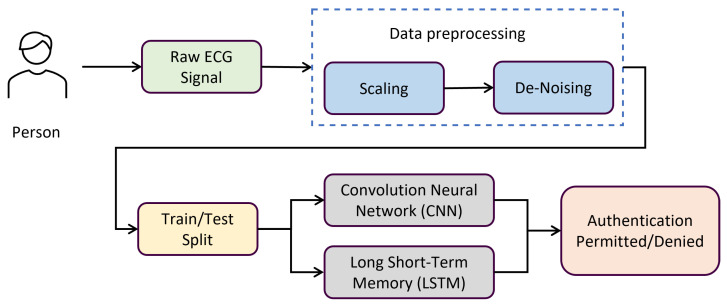
Block diagram proposed methodology.

**Figure 4 diagnostics-13-00439-f004:**
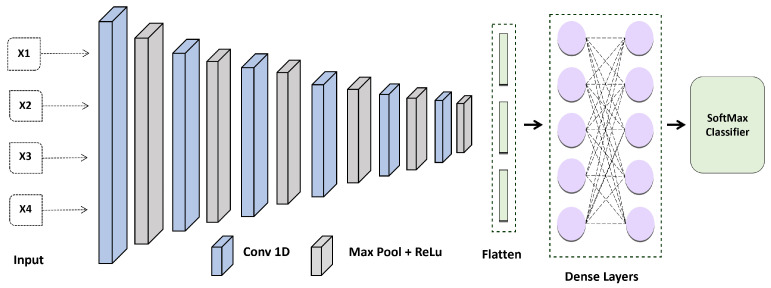
Proposed Convolution Neural Network Model for ECG-based Authentication.

**Figure 5 diagnostics-13-00439-f005:**
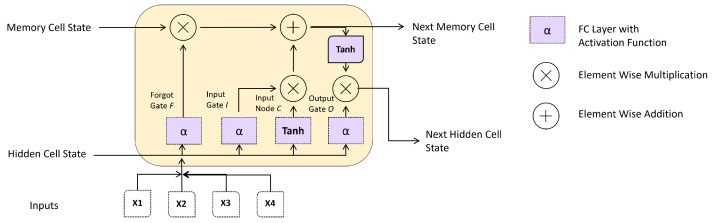
Main components Long Short-Term Memory (LSTM) Cell.

**Figure 6 diagnostics-13-00439-f006:**
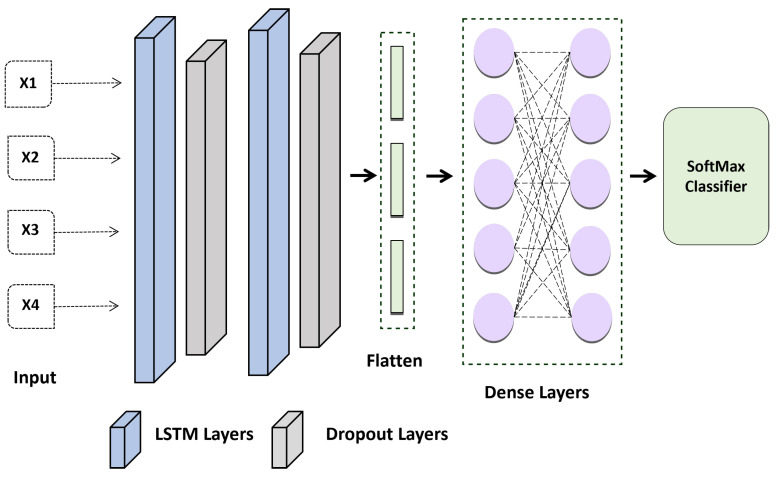
Proposed LSTM Model for ECG-based Authentication.

**Figure 7 diagnostics-13-00439-f007:**
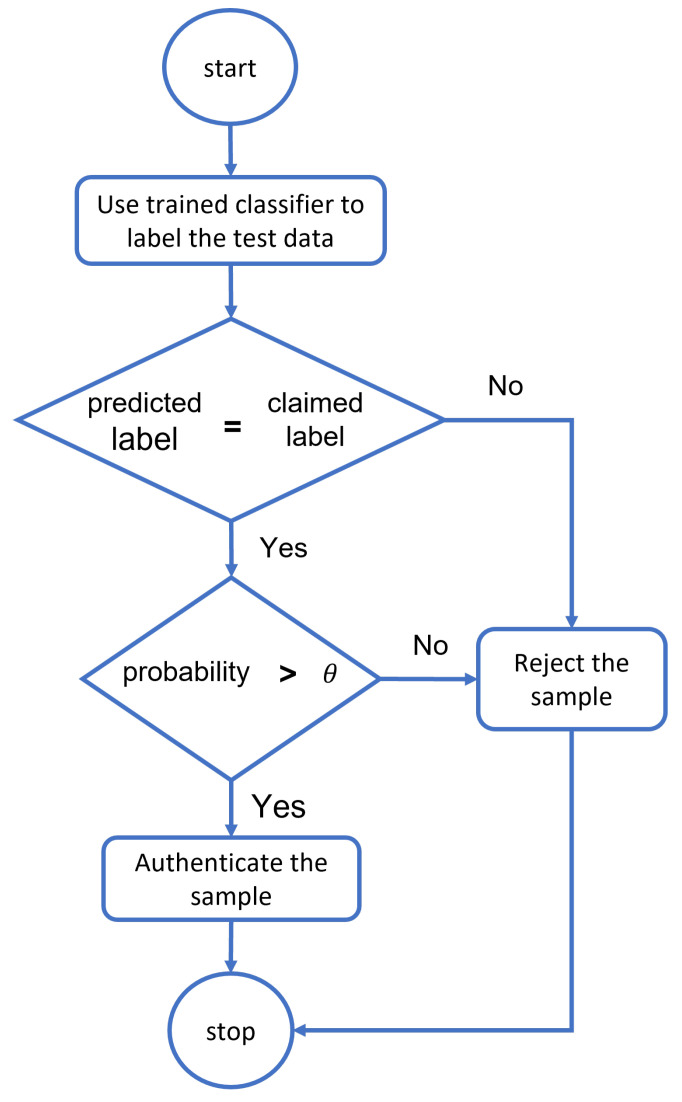
The flowchart of threshold-based classifiers for authentication.

**Figure 8 diagnostics-13-00439-f008:**
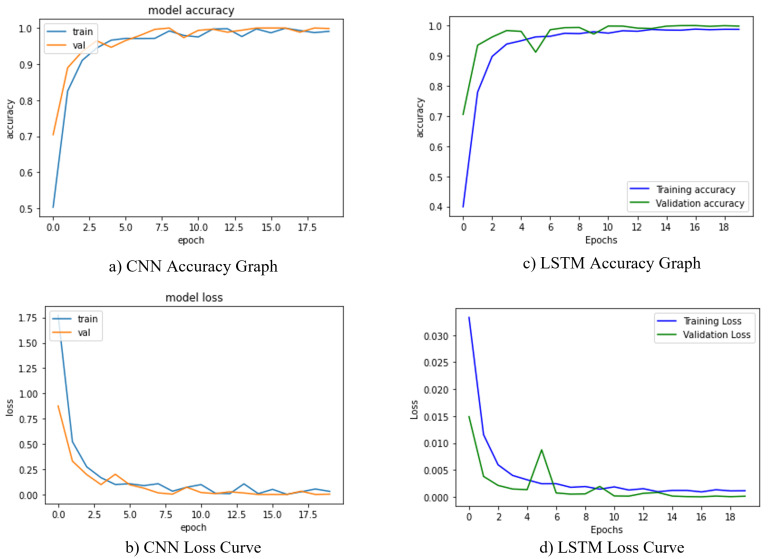
Training and validation loss curves of the proposed model. (**a**,**b**) shows the accuracy and loss curves for CNN model. (**c**,**d**) represents the accuracy and loss curves for the LSTM model.

**Figure 9 diagnostics-13-00439-f009:**
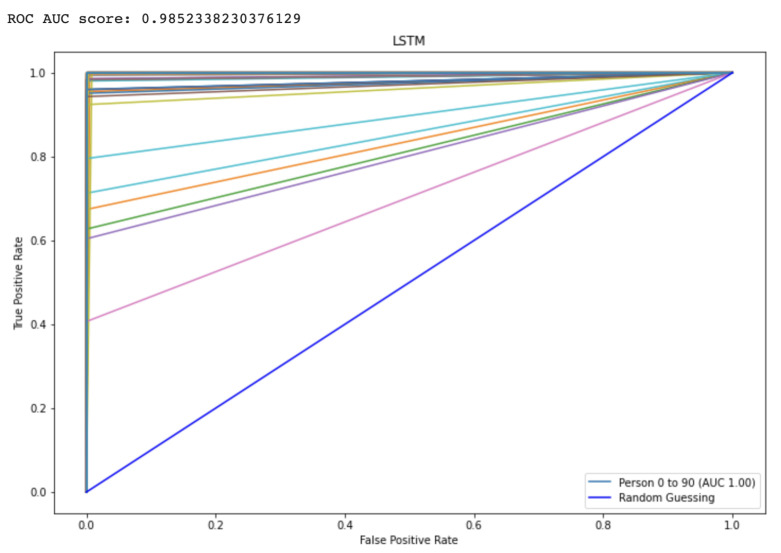
Represents the area under the curve score for proposed LSTM model.

**Figure 10 diagnostics-13-00439-f010:**
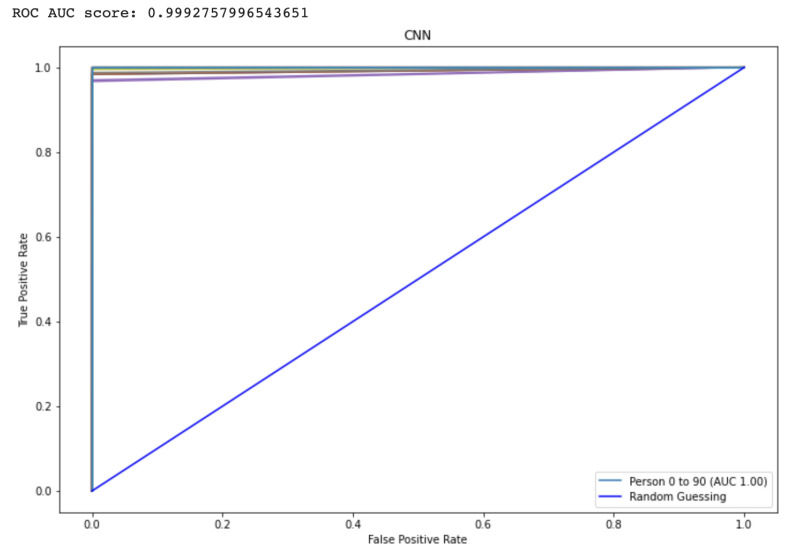
Represents the area under the curve score for the proposed CNN model.

**Table 1 diagnostics-13-00439-t001:** Two scenarios for the membership probability according to similarity to each class distribution in a 5-class problem.

Probabilities	1	2	3	4	5
Scenario A	0.91	0.01	0.01	0.01	0.01
Scenario A	0.45	0.33	0.15	0.01	0.01

**Table 2 diagnostics-13-00439-t002:** Hyper-parameters settings for training.

No.	Hyperparameters	Settings
1	Loss Function	Binary Cross-entropy
2	Optimizer	Adam Optimization
3	Learning rate	0.001
4	Epochs	20
5	Batch size	32
6	Callbacks	Model Checkpoint

**Table 3 diagnostics-13-00439-t003:** Comparison of the proposed method with other SOTA methods.

Metric	Models	Accuracy (%)
Lynn et al. [12]	Bidirectional GRU	0.985
Karpinski et al. [20]	Auto-encoder	0.892
Lee et al. [24]	Ensemble	0.984
Akeem et al. [26]	Decision Tree	0.920
Proposed Model (Ours)	CNN and LSTM	0.983 & 0.996

**Table 4 diagnostics-13-00439-t004:** Metrics evaluation results of proposed CNN and LSTM models.

Metric	CNN (%)	LSTM (%)
Accuracy	0.983	0.996
Recall	0.999	1
Precision	0.992	0.997
F1-Score	0.999	1
AUC Score	0.992	98.52

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
