# Peer review of "Electrocardiogram (ECG)-Based User Authentication Using Deep Learning Algorithms"

_diagnostics, 2023, doi:10.3390/diagnostics13030439_

Round 1

Reviewer 1 Report

This paper is in line with the scope of this journal with some significant contributions, but I am concerned about the following questions:

l   I think that not all symbols are defined for equations, especially in the part where a novel method is presented.

l  Expend conclusion to include details regarding the future work.

l   There are some technical and English language errors, please read the manuscript carefully and revise.

l  Please remove “we” from the manuscript and instead use the proposed system.

Reviewer 2 Report

Main novelty of Long Short Term Memory (LSTM) neural network should be illustrated exactly in Abstract and Section 1.

Authors ignored main Section of literature review in this paper. Authors should add some discussions on existing relevant studies as follows:

1- EDITH: ECG biometrics aided by Deep learning for reliable Individual auTHentication

2- Event-driven IoT architecture for data analysis of reliable healthcare application using complex event processing 

3- BAED: A secured biometric authentication system using ECG signal based on deep learning techniques

4- An elderly health monitoring system based on biological and behavioral indicators in internet of things

Also, existing Feature Engineering section is unclear. Please explain exactly the main novelty of feature selection approach with details.

The existing flowchart of threshold-based multi-class classifiers for authentication as Figure 8 should be more clarified.

Reviewer 3 Report

Comments and Final Decision about This research
paper above mentioned

1 - About the title: ECG-Based User Authentication Using Deep Learning Architectures.
Comments:
The reviewer suggests the following title: “ElectroCardioGram (ECG) -Based User
Authentication using Deep learning Algorithms”.

2 - About the Abstract:
Comments:
The abstract can be improved (the authors must be highlighted their methodology and the
essential of their contributions and findings). Adequate methodology needed to be mentioned in
the abstract focusing on deep learning simulations). The authors should state the results and
fundings briefly in the abstract.

3 - About the Keywords:
Comments:
The reviewer suggests the following keywords: “User Authentication,
ElectroCardioGram (ECG), Deep Learning Algorithms, LSTM and PTB database,
Telehealth System, CNN & LSTM Training and validation”.

4 - About the Introduction:
Comments:
The authors should enhance the introduction. The reviewer suggests that a summarized
table involving the main problems indexed in the topic around the world during the last decades.
Also, the authors should mention similar case studies especially in specific country or area. From
the 40 references involved in this article only 24 references have been cited in the introduction.

5 - About Materials and Methods:
Comments:
This section must be more organized, the authors should cleary highlight the materials used
in this article and the methods adopted. The reviewer found at first a short paragraph in which the
authors explained the deepl learning techniques. After that, we found the first subtitle (2.1.
Convolution Neural Networks (CNN)). In this sub-section the authors should explain CNN by
using figure vusializing the convolutional layers to reflect learning features based on data. The
same way to the second sub-section (2.2. Long Short-Term Memory Networks).
The third sub-section (2.3. Hyperparameters and Loss Function) considered as method
for tackling the problem as the authors written. This paragraph as methods is very short and needed
to be improved.
As fourth sub-section the reviewer suggests that the software applied must be involved
(page 9, the authors described one software (Waveform Database Software Package (WFDB)).
Also, in this sub-section the authors should mention the flowchart of the methodology used to
perform the simulations using any software.
In the section (3. Data Preprocessing and Experimental Work), the reviewer found that
this name of section must be changed to (3. Data Collection and Simulation using Deep learning
algorithms).
The five sub-sections must be revised respectively.
The figure 3 must be improved explicitly and must be designed up to down according to
the books.
The figure 4 must be improved explicitly and must be designed up to down according to
the books.

The figure 5 must be improved explicitly and must be designed up to down according to
the books.
The figure 6 must be improved explicitly and must be designed up to down according to
the books (Permitted no any link with the remaining flowchart).
The figure 7 must be improved (each component must be identified).
The figure 8 must be improved and combined with the figure 6.

6 – Results and Discussion:
Comments:
In this section (Results and Discussion), the authors should separately give one paragraph
for Results and another for Discussion. The first one dealing with the main results found based on
the simulations using Deep learning algorithms (trainings and validations). This second one
dealing with the performance of these tools compared with what we found in the recent literature
review.
The figure 9 concerning training and validation loss curves of the proposed model (in figure
9d why the loss is very little (maximum 0.030)?).
In the figure 10 concerning the AUC curves for CNN model and LSTM model not clear
and not representative.
In the figure 11 concerning the confusion matrix method used to evaluate the performance
of the proposed methods (this figure not clear and no any representative results).

7 – Conclusion and Future Work:
Comments:
The authors should focus on the main findings in the conclusion. In addition, at the end of
conclusion, they should develop the essential of what we can do in the future as recommendations
and perspectives such as how to involve SMV machine learning algorithm combined with Deep
learning algorithms cited in this article. This section must be improved.
9 – General remarks:
Comments:
1 – All figures must be improved;
2 – Problem identification must be added in the Introduction section;
3 – Methodology flowchart must be modified in materials and methods;
4 – Processing and simulation (Training and validation) must be clearly described in materials and
methods;
5 – Adequate methodology flowchart taking into consideration deep learning algorithms must be
involved in the above Flowchart in materials and methods section;
6 – All figures must be involved;
7 – Discussion, conclusions, and future work must be improved and justify how to involve deep
learning using Electrocardiogram and Telehealth System database.
10 - Final decision about this paper: ECG-Based User Authentication Using Deep Learning
Architectures.
• I propose to the Diagnostics Journal to accept this research paper after major revision taking into
consideration the above comments for each section.

Round 2

Reviewer 3 Report

Accepted in the present form.